# Action is in the Eye of the Beholder: Eye-gaze Driven Model for Spatio-Temporal Action Localization

**Nataliya Shapovalova**[*]     **Michalis Raptis**[†]     **Leonid Sigal**[‡]     **Greg Mori**[*]

[*]Simon Fraser University     [†]Comcast     [‡]Disney Research

{nshapova,mori}@cs.sfu.ca   mraptis@cable.comcast.com   lsigal@disneyresearch.com

## Abstract

We propose a weakly-supervised structured learning approach for recognition and spatio-temporal localization of actions in video. As part of the proposed approach, we develop a generalization of the Max-Path search algorithm which allows us to efficiently search over a structured space of multiple spatio-temporal paths while also incorporating context information into the model. Instead of using spatial annotations in the form of bounding boxes to guide the latent model during training, we utilize human gaze data in the form of a weak supervisory signal. This is achieved by incorporating eye gaze, along with the classification, into the structured loss within the latent SVM learning framework. Experiments on a challenging benchmark dataset, UCF-Sports, show that our model is more accurate, in terms of classification, and achieves state-of-the-art results in localization. In addition, our model can produce top-down saliency maps conditioned on the classification label and localized latent paths.

## 1   Introduction

Structured prediction models for action recognition and localization are emerging as prominent alternatives to more traditional holistic bag-of-words (BoW) representations. The obvious advantage of such models is the ability to localize, spatially and temporally, an action (and actors) in potentially long and complex scenes with multiple subjects. Early alternatives [3, 7, 14, 27] address this challenge using sub-volume search, however, this implicitly assumes that the action and actor(s) are static within the frame. More recently, [9] and [18, 19] propose figure-centric approaches that can *track* an actor by searching over the space of spatio-temporal paths in video [19] and by incorporating person detection into the formulation [9]. However, all successful localization methods, to date, require spatial annotations in the form of partial poses [13], bounding boxes [9, 19] or pixel level segmentations [7] for learning. Obtaining such annotations is both time consuming and unnatural; often it is not easy for a human to decide which spatio-temporal segment corresponds to an action.

One alternative is to proceed in a purely unsupervised manner and try to mine for most *discriminant* portions of the video for classification [2]. However, this often results in overfitting due to the relatively small and constrained nature of the datasets, as discriminant portions of the video, in training, may correspond to regions of background and be unrelated to the motion of interest (e.g., grass may be highly discriminative for "kicking" action because in the training set most instances come from soccer, but clearly "kicking" can occur on nearly any surface). Bottom-up perceptual *saliency*, computed from eye-gaze of observers (obtained using an eye tracker), has recently been introduced as another promising alternative to annotation and supervision [11, 21]. It has been shown that traditional BoW models computed over the salient regions of the video result in superior performance, compared to dense sampling of descriptors. However, this comes at expense of losing ability to localize actions. Bottom-up saliency models usually respond to numerous unrelated low-level stimuli [25](e.g., textured cluttered backgrounds, large motion gradients from subjects irrelevant to the action, etc.) which often fall outside the region of the action (and can confuse classifiers).

In this paper we posit that a good spatio-temporal model for action recognition and localization should have three key properties: (1) be figure-centric, to allow for subject and/or camera motion, (2) discriminative, to facilitate classification and localization, and (3) perceptually semantic, to mitigate overfitting to accidental statistical regularities in a training set. To avoid reliance on spatial annotation of actors we utilize human gaze data (collected by having observers view corresponding videos [11]) as weak supervision in learning[1]. Note that such weak annotation is more natural, effortless (from the point of view of an annotator) and can be done in real-time. By design, gaze gives perceptually semantic interest regions, however, while semantic, gaze, much like bottom-up saliency, is not necessarily discriminative. Fig. 1(b) shows that while for some (typically fast) actions like "diving", gaze may be well aligned with the actor and hence discriminative, for others, like "golf" and "horse riding", gaze may either drift to salient but non discriminant regions (the ball), or simply fall on background regions that are prominent or of intrinsic aesthetic value to the observer.

To deal with complexities of the search and ambiguities in the weak-supervision, given by gaze, we formulate our model in a max-margin framework where we attempt to infer latent smooth spatio-temporal path(s) through the video that simultaneously maximize classification accuracy and pass through regions of high gaze concentration. During learning, this objective is encouraged in the latent Structural SVM [26] formulation through a real-valued loss that penalizes misclassification and, for correctly classified instances, misalignment with salient regions induced by the gaze. In addition to classification and localization, we show that our model can provide top-down action-specific saliency by predicting distribution over gaze conditioned on the action label and inferred spatio-temporal path. Having less (annotation) information available at training time, our model shows state-of-the art classification and localization accuracy on the UCF-Sports dataset and is the first, to our knowledge, to propose top-down saliency for action classification task.

## 2   Related works

**Action recognition:** The literature on vision-based action recognition is too vast. Here we focus on the most relevant approaches and point the reader to recent surveys [20, 24] for a more complete overview. The most prominent action recognition models to date utilize visual BoW representations [10, 22] and extensions [8, 15]. Such holistic models have proven to be surprisingly good at recognition, but are, by design, incapable of spatial or temporal localization of actions.

**Saliency and eye gaze:** Work in cognitive science suggests that control inputs to the attention mechanism can be grouped into two categories: stimulus-driven (bottom-up) and goal-driven (top-down) [4]. Recent work in action recognition [11, 21] look at bottom-up saliency as a way to sparsify descriptors and to bias BoW representations towards more salient portions of the video. In [11] and [21] multiple subjects were tasked with viewing videos while their gaze was recorded. A saliency model is then trained to predict the gaze and is used to either prune or weight the descriptors. However, the proposed saliency-based sampling is purely bottom-up, and still lacks ability to localize actions in either space or time[2]. In contrast, our model is designed with spatio-temporal localization in mind and uses gaze data as weak supervision during learning. In [16] and [17] authors use "objectness" saliency operator and person detector as weak supervision respectively, however, in both cases the saliency is bottom-up and task independent. The *top-down* discriminative saliency, based on distribution of gaze, in our approach, allows our model to focus on *perceptually salient* regions that are also *discriminative*. Similar in spirit, in [5] gaze and action labels are simultaneously inferred in ego-centric action recognition setting. While conceptually similar, the model in [5] is significantly different both in terms of formulation and use. The model [5] is generative and relies on existence of object detectors.

**Sub-volume search:** Spatio-temporal localization of actions is a difficult task, largely due to the computational complexity of search involved. One way to alleviate this computational complexity is to model the action as an axis aligned rectangular 3D volume. This allows spatio-temporal search to be formulated efficiently using convolutions in the Fourier [3] or Clifford Fourier [14] domain. In [28] an efficient spatio-temporal branch-and-bound approach was proposed as alternative. However, the assumption of single fixed axis aligned volumetric representation is limiting and only applicable

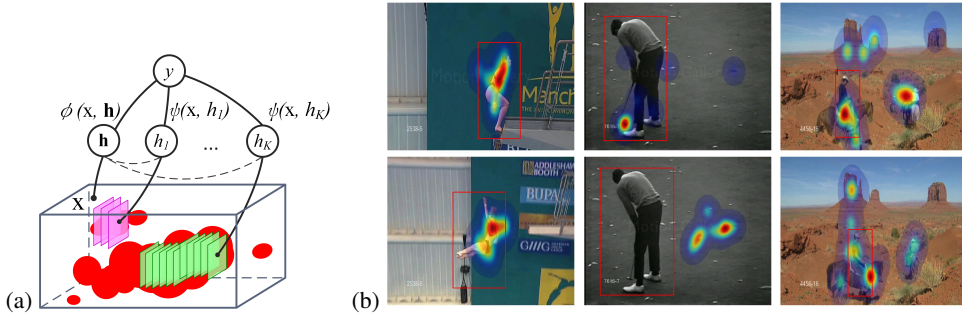

Figure 1: **Graphical model representation** is illustrated in (a). Term $\phi(\mathbf{x}, \mathbf{h})$ captures information about context (all the video excluding regions defined by latent variables $\mathbf{h}$); terms $\psi(\mathbf{x}, h_i)$ capture information about latent regions. Inferred latent regions should be discriminative and match high density regions of eye gaze data. In (b) ground truth eye gaze density, computed from fixations of multiple subjects, is overlaid over images from sequences of 3 different action classes (see Sect. 1).

for well defined and relatively static actions. In [7] an extension to multiple sub-volumes that model parts of the action is proposed and amounts to a spatio-temporal part-based (pictorial structure) model. While part-based model of [7] allows for greater flexibility, the remaining axis-aligned nature of part sub-volumes is still largely appropriate for recognition in scenarios where camera and subject are relatively static. This constraint is slightly relaxed in [12] where a part-based model built on dense trajectory clustering is proposed. However, [12] relies on sophisticated pre-processing which requires building long feature trajectories over time, which is difficult to do for fast motions or less textured regions.

Most closely related approaches to our work come from [9, 18, 19]. In [18] Tran and Yuan show that a rectangular axis-aligned volume constraint can be relaxed by efficiently searching over the space of smooth paths within the spatio-temporal volume. The resulting Max-Path algorithm is applied to object tracking in video. In [19] this approach is further extended by incorporating Max-Path inference into a max-margin structured output learning framework, resulting in an approach capable of localizing actions. We generalize Max-Path idea by allowing multiple smooth paths and context within a latent max-margin structured output learning. In addition, our model is trained to simultaneously localize and classify actions. Alternatively, [9] uses latent SVM to jointly detect an actor and recognize actions. In practice, [9] relies on human detection for both inference and learning and only sub-set of frames can be localized due to the choice of the features (HOG3D). Similarly, [2] relies on person detection and distributed partial pose representation, in the form of poselets, to build a spatio-temporal graph for action recognition and localization. We want to stress that [2, 9, 18, 19] require bounding box annotations for actors in learning. In contrast, we focus on weaker and more natural source of data – gaze, to formulate our learning criteria.

## 3 Recognizing and Localizing Actions in Videos

Our goal is to learn a model that can jointly localize and classify human actions in video. This problem is often tackled in the same manner as object recognition and localization in images. However, extension to a temporal domain comes with many challenges. The core challenges we address are: (i) dealing with a motion of the actor within the frame, resulting from camera or actor's own motion in the world; (ii) complexity of the resulting spatio-temporal search, that needs to search over the space of temporal paths; (iii) ability to model coarse temporal progression of the action and action context, and (iv) learning in absence of direct annotations for actor(s) position within the frame.

To this end, we propose a model that has the ability to localize temporally and spatially discriminative regions of the video and encode the context in which these regions occur. The output of the model indicates the absence or presence of a particular action in the video sequence while simultaneously extracting the most discriminative and perceptually salient spatio-temporal video regions. During the training phase, the selection of these regions is implicitly driven by eye gaze fixations collected by a sample of viewers. As a consequence, our model is able to perform top-down video saliency detection conditioned on the performed action and localized action region.

# 1 Model Formulation

Given a set of video sequences $\{\mathbf{x}_1, \ldots, \mathbf{x}_n\} \subset \mathcal{X}$ and their associated labels $\{y_1, \ldots, y_n\}$, with $y_i \in \{-1, 1\}$, our purpose is to learn a mapping $f : \mathcal{X} \rightarrow \{-1, 1\}$. Additionally, we introduce auxiliary *latent* variables $\{\mathbf{h}_1, \ldots, \mathbf{h}_n\}$, where $\mathbf{h}_i = \{h_{i1}, \ldots, h_{iK}\}$ and $h_{ik} \in \emptyset \cup \{(l^j, t^j, r^j, b^j)_{j=T_s}^{T_e}\}$ denotes the left, top, right and bottom coordinates of spatio-temporal paths of bounding boxes that are defined from frame $T_s$ up to $T_e$. The latent variables $\mathbf{h}$ specify the spatio-temporal regions selected by our model. Our function is then defined $y_{\mathbf{x}}^*(w) = f(\mathbf{x}; w)$, where

$$(y_{\mathbf{x}}^*(w), \mathbf{h}_{\mathbf{x}}^*(w)) = \underset{(y,\mathbf{h}) \in \{-1,1\} \times \mathcal{H}}{\operatorname{argmax}} F(\mathbf{x}, y, \mathbf{h}; w), \quad F(\mathbf{x}, y, \mathbf{h}; w) = w^T \Psi(\mathbf{x}, y, \mathbf{h}), \quad (1)$$

$w$ is a parameter of the model, and $\Psi(\mathbf{x}, y, \mathbf{h}) \in \mathbb{R}^d$ is a joint feature map. Video sequences in which the action of interest is absent are treated as zero vectors in the Hilbert space induced by the feature map $\Psi$ similar to [1]. Whereas, the corresponding feature map of videos where the action of interest is present is being decomposed into two components: a) the *latent regions* and b) *context regions*. As a consequence, the scoring function is written:

$$F(\mathbf{x}, y = 1, \mathbf{h}; w) = w^T \Psi(\mathbf{x}, y = 1, \mathbf{h}) = w_0^T \phi(\mathbf{x}, \mathbf{h}) + \sum_{k=1}^{K} w_k^T \psi(\mathbf{x}, h_k) + b \quad (2)$$

where $K$ is the number of latent regions of the action model and $b$ is the bias term. A graphical representation of the model is illustrated in Fig. 1(a).

**Latent regions potential** $w_k^T \psi(\mathbf{x}, h_k)$: This potential function measures the compatibility of latent spatio-temporal region $h_k$ with the action model. More specifically, $\psi(\mathbf{x}, h_k)$ returns the sum of normalized BoW histograms extracted from the bounding box defined by the latent variable $h_k = (l^j, t^j, r^j, b^j)_{j=T_s}^{T_e}$ at each corresponding frame.

**Context potential** $w_0^T \phi(\mathbf{x}, \mathbf{h})$: We define context as the entire video sequence excluding the latent regions; our aim is to capture any information that is not directly produced by the appearance and motion of the actor. The characteristics of the context are encoded in $\phi(\mathbf{x}, \mathbf{h})$ as a sum of normalized BoW histograms at each frame of the video excluding the regions indicated by latent variables $\mathbf{h}$.

Many action recognition scoring functions recently proposed [9, 12, 16] include the response of a global BoW statistical representation of the entire video. While such formulations are simpler, since the response of the global representation is independent from the selection of the latent variables, they are also somewhat unsatisfactory from the modeling point of view. First, the visual information that corresponds to the latent region of interest implicitly gets to be counted twice. Second, it is impossible to decouple and analyze importance of foreground and contextual information separately.

# 2 Inference

Given the model parameters $w$ and an unseen video $\mathbf{x}$ our goal is to infer the binary action label $y^*$ as well as the location of latent regions $\mathbf{h}^*$ (Eq. 1). The scoring function for the case of $y = -1$ is equal to zero due to the trivial zero vector feature map (Sect. 1). However, estimating the optimal value of the scoring function for the case of $y = 1$ involves the maximization over the latent variables. The search space over even a single spatio-temporal path (non-smooth) of variable size bounding boxes in a video sequence of width $M$, height $N$ and length $T$ is exponential: $O(MN)^{2T}$. Therefore, we restrict the search space by introducing a number of assumptions. We constraint the search space to smooth spatio-temporal paths[3] of fixed size bounding boxes [18]. These constraints allows the inference of the optimal latent variables for a single region using dynamic programming, similarly to *Max-Path* algorithm proposed by Tran and Yuan [18].

Algorithm 1 summarizes the process of dynamic programming considering both the context and the latent region contributions. The time and space complexity of this algorithm is $O(MNT)$. However, without introducing further constraints on the latent variables, the extension of this forward message passing procedure to multiple latent regions results in an exponential, in the number of regions, algorithm because of the implicit dependency of the latent variables through the context

**Algorithm 1** `MaxCPath`: Inference of Single Latent Region with Context

---

1: **Input** : $R(t)$: the context local response without the presence of bounding box,
   $\qquad Q_0(u, v, t)$: the context local response excluding the bounding box at location $(u, v)$,
   $\qquad Q_1(u, v, t)$: the latent region local response
2: **Output** : $S(t)$: score of the best path till frame $t$, $L(t)$: end point of the best path till $t$,
   $\qquad P(u, v, t)$: the best path record for tracing back
3: Initialize $S^* = - \inf, S(u, v, 0) = -inf, \forall u, v, l^* = null$
4: **for** $t \leftarrow 1$ **to** $T$ **do** // *Forward Process, Backward Process*: $t \leftarrow T$ **to** 1
5: $\quad$ **for each** $(u, v) \in [1..M] \times [1..N]$ **do**
6: $\qquad (u_0, v_0) \leftarrow \operatorname{argmax}_{(u', v') \in \mathrm{Nb}(u,v)} S(u', v', t-1)$
7: $\qquad$ **if** $S(u_0, v_0, t-1) > \sum_{i=1}^{T} R(i)$ **then**
8: $\qquad\quad S(u, v, t) \leftarrow S(u_0, v_0, t-1) + Q_0(u, v, t) + Q_1(u, v, t) - R(t)$
9: $\qquad\quad P(u, v, t) \leftarrow (u_0, v_0, t-1)$
10: $\qquad$ **else**
11: $\qquad\quad S(u, v, t) \leftarrow Q_0(u, v, t) + Q_1(u, v, t) + \sum_{i=1}^{T} R(i) - R(t)$
12: $\qquad$ **end if**
13: $\qquad$ **if** $S(u, v, t) > S^*$ **then**
14: $\qquad\quad S^* \leftarrow S(u, v, t)$ and $l^* \leftarrow (u, v, t)$
15: $\qquad$ **end if**
16: $\quad$ **end for**
17: $\quad S(t) \leftarrow S^*$ and $L(t) \leftarrow l^*$
18: **end for**

---

**Algorithm 2** Inference: Two Latent Region with Context

---

1: **Input** : $R(t)$: the context local response without the presence of bounding box, $Q_0(u, v, t)$: the context local response excluding the bounding box at location $(u, v)$, $Q_1(u, v, t)$: the latent region local response of the first latent region, $Q_2(u, v, t)$: the latent region local response of the second latent region.
2: **Output** : $S^*$: the maximum score of the inference, $h_1, h_2$: first and second latent regions
3: Initialize $S^* = - \inf, t^* = null$
4: $(S_1, L_1, P_1) \leftarrow MaxCPath - Forward(R, Q_0, Q_1)$
5: $(S_2, L_2, P_2) \leftarrow MaxCPath - Backward(R, Q_0, Q_2)$
6: **for** $t \leftarrow 1$ **to** $T - 1$ **do**
7: $\quad S \leftarrow S_1(t) + S_2(t+1) - \sum_{i=1}^{T} R(i)$
8: $\quad$ **if** $S > S^*$ **then**
9: $\qquad S^* \leftarrow S$ and $t^* \leftarrow t$
10: $\quad$ **end if**
11: **end for**
12: $h_1 \leftarrow traceBackward(P_1, L_1(t^*))$
13: $h_2 \leftarrow traceForward(P_2, L_2(t^* + 1))$

---

term. Incorporating temporal ordering constraints between the $K$ latent regions leads to a polynomial time algorithm. More specifically, the optimal scoring function can be inferred by enumerating all potential end locations of each latent region and executing independently Algorithm 1 at each interval in $O(MNT^K)$. For the special case of $K = 2$, we derive a forward/backward message process that remains linear in the size of video volume: $O(MNT)$; see summary in Algorithm 2. In our experimental validation a model with 2 latent regions proved to be sufficiently expressive.

## 3 Learning Framework

Identifying the spatio-temporal regions of the video sequences that will enable our model to detect human action is a challenging optimization problem. While the introduction of latent variables in discriminative models [6, 9, 12, 13, 23, 26] is natural for many applications (e.g., modeling body parts) and has also offered excellent performance [6], it also lead to training formulations with non-convex functions. In our training formulation we adopt the large-margin latent structured output learning [26], however we also introduce a loss function that weakly supervises the selection of latent variables based on human gaze information. Our training set of videos $\{\mathbf{x}_1, \ldots, \mathbf{x}_n\}$ along with their action labels $\{y_1, \ldots, y_n\}$ contains 2D fixation points (sampled at much higher frequency than the video frame rate) of 16 subjects observing the videos [11]. We transform these measurements using kernel density estimation with Gaussian kernel (with bandwidth set to the visual angle span of $2°$) to a probability density function of gaze $\mathbf{g}_i = \{g_i^1, \ldots, g_i^{T_i}\}$ at each frame of video $\mathbf{x}_i$. Following the Latent Structural SVM formulation [26], our learning takes the following form:

$$\min_{w, \xi \geq 0} \frac{1}{2} \|w\|^2 + C \sum_{i=1}^{n} \xi_i \qquad (3)$$

$$\max_{\mathbf{h}_i' \in \mathcal{H}} w^T \Psi(\mathbf{x}_i, y_i, \mathbf{h}_i') - w^T \Psi(\mathbf{x}_i, \hat{y}_i, \hat{\mathbf{h}}_i) \geq \Delta(y_i, \mathbf{g}_i, \hat{y}_i, \hat{\mathbf{h}}_i) - \xi_i, \quad \forall \hat{y}_i \in \{-1, 1\}, \forall \hat{\mathbf{h}}_i \in \mathcal{H},$$

where $\Delta(y_i, \mathbf{g}_i, \hat{y}_i, \hat{\mathbf{h}}_i) \geq 0$ is an asymmetric loss function encoding the cost of an incorrect action label prediction but also of mislocalization of the eye gaze. The loss function is defined as follows:

$$\Delta(y_i, \mathbf{g}_i, \hat{y}_i, \hat{\mathbf{h}}_i) = \begin{cases} 1 - \frac{1}{K}\sum_{k=1}^{K} \delta(\mathbf{g}_i, \hat{h}_{ik}) & \text{if } y_i = \hat{y}_i = 1, \\ 1 - \frac{1}{2}(y_i\hat{y}_i + 1) & \text{otherwise.} \end{cases} \qquad (4)$$

$\delta(\mathbf{g}_i, \hat{h}_{ik})$ indicates the minimum overlap of $\hat{h}_{ik}$ and a given eye gaze $\mathbf{g}_i$ map over a frame:

$$\delta(\mathbf{g}_i, \hat{h}_{ik}) = \min_j \delta_p(b_{ik}^j, g_i^j), \quad T_{s,k} \leq j \leq T_{e,k}, \qquad (5)$$

$$\delta_p(b_{ik}^j, g_i^j) = \begin{cases} 1 & \text{if } \sum_{b_{ik}^j} g_i^j \geq r, \quad 0 < r < 1, \\ \frac{1}{r}\sum_{b_{ik}^j} g_i^j & \text{otherwise,} \end{cases} \qquad (6)$$

where $b_{ik}^j$ is the bounding box at frame $j$ of the $k$-th latent region in the $\mathbf{x}_i$ video. The parameter $r$ regulates the minimum amount of eye gaze "mass" that should be enclosed by each bounding box. The loss function can be easily incorporated in Algorithm 1 during the loss-augmented inference.

## 4   Gaze Prediction

Our model is based on the core assumption that a subset of perceptually salient regions of a video, encoded by the gaze map, share discriminative idiosyncrasies useful for human action classification. The loss function dictating the learning process enables the model's parameter (*i.e* , $w$) to encode this notion into our model[4]. Assuming our assumption holds in practice, we can use selected latent regions for prediction of top-down saliency within the latent region. We do so by regressing the amount of eye gaze (probability density map over gaze) on a fixed grid, inside each bounding box of the latent regions, by conditioning on low level features that construct the feature map $\psi_i$ and the action label. In this way the latent regions select consistent salient portions of videos using top-down knowledge about the action, and image content modulates the saliency prediction within that region.

Given the training data gaze $\mathbf{g}$ and the corresponding inferred latent variables $\mathbf{h}$, we learn a linear regression, per action class, that maps augmented feature representation of the extracted bounding boxes, of each latent region, to a coarse description of the corresponding gaze distribution. Each bounding box is divided into a $4 \times 4$ grid and a BoW representation for each cell is computed; augmented feature is constructed by concatenating these histograms. Similarly, the human gaze is summarized by a 16 dimension vector by accumulating gaze density at each cell over a $4 \times 4$ grid. For visualization, we further smooth the predictions to obtain a continuous and smooth gaze density over the latent regions. We find our top-down saliency predictions to be quite good (see Sect. 5) in most cases which experimentally validated our initial model assumption.

## 5   Experiments

We evaluate our model on the UCF-Sports dataset presented in [14]. The dataset contains 150 videos extracted from broadcast television channels and includes 10 different action classes. The dataset includes annotation of action classes as well as bounding boxes around the person of interest (which we ignore for training but use to measure localization performance). We follow the evaluation setup defined in the of Lan *et al.* [9] and split the dataset into 103 training and 47 test samples. We employ the eye gaze data made available by Mathe and Sminchisescu [11]. The data captures eye movements of 16 subjects while they were watching the video clips from the dataset. The eye gaze data are represented with a probability density function (Sect. 4).

**Data representation:** We extract HoG, HoF, and HoMB descriptors [12] at a dense spatio-temporal grid and at 4 different scales. These descriptors are clustered into 3 vocabularies of 500, 500, 300 sizes correspondingly. For the baseline experiments, we use $\ell_1$-normalized histogram representation. For the potentials described in Sect. 1, we represent latent regions/context with the sum of per-frame normalized histograms. Per-frame normalization, as opposed to global normalization over the spatio-temporal region, allows us to aggregate scores iteratively in Algorithm 1.

**Baselines:**   We compare our model to several baseline methods. All our baselines are trained with linear SVM, to make them comparable to our linear model, and use the same feature representation

| | Model | Accuracy | | Localization | |
|---|---|---|---|---|---|
| **Baselines** | Global BoW | 64.29 | | N/A | |
| | BoW with SS | 65.95 | | N/A | |
| | BoW with TS | 69.64 | | N/A | |
| | | # of Latent Regions | | | |
| | | $K=1$ | $K=2$ | $K=1$ | $K=2$ |
| **Our Model** | Region | 77.98 | **82.14** | 26.4 | 20.8 |
| | Region+Context | 77.62 | 81.31 | **32.3** | 29.3 |
| | Region+Global | 76.79 | 80.71 | 29.6 | 30.4 |
| **State-of-the-art** | Lan *et al.* [9] | 73.1 | | 27.8 | |
| | Tran and Yuan [19] | N/A | | 54.3* | |
| | Shapovalova *et al.* [16] | 75.3 | | N/A | |
| | Raptis *et al.* [12] | 79.4 | | N/A | |

Table 1: **Action classification and localization results.** Our model significantly outperforms the baselines and most of the State-of-the-art results (see text for discussion). *Note that the average localization score is calculated based only on three classes reported in [19].

as described above. We report performance of three baselines: (1) **Global BoW**, where video is represented with just one histogram and all the temporal-spatial structure is discarded. (2) **BoW with spatial split (SS)**, where video is divided by a $2 \times 2$ spatial grid and parts in order to capture spatial structure. (3) **BoW with temporal split (TS)**, where the video is divided into 2 consecutive temporal segments. This setup allows the capture of the basic temporal structure of human action.

**Our model:** We evaluate three different variants of our model, which we call **Region**, **Region+Global**, and **Region+Context**. **Region**: includes only the latent regions, the potentials $\psi$ from our scoring function in Eq. 1, and ignores the context features $\phi$. **Region+Global**: the context potential $\phi$ is replaced with a Global BoW, like in our first baseline. **Region+Context**: represents our full model from the Eq. 1. We test all our models with one and two latent regions.

**Action classification and localization:** Results of action classification are summarized in Table 1. We train a model for each action separately in a standard one-vs-all framework. Table 1 shows that all our models outperform the BoW baselines and the results of Lan *et al.* [9] and Shapovalova *et al.* [16]. The **Region** and **Region+Context** models with two latent regions demonstrate superior performance compared to Raptis *et al.* [12]. Our model with 1 latent region performs slightly worse then model of Raptis *et al.* [12], however note that [12] used non-linear SVM with $\chi^2$ kernel and 4 regions, while we work with linear SVM only. Further, we can clearly see that having 2 latent regions is beneficial, and improves the classification performance by roughly $4\%$. The addition of **Global** BoW marginally decreases the performance, due to, we believe, over counting of image evidence and hence overfitting. **Context** does not improve classification, but does improve localization.

We perform action localization by following the evaluation procedure of [9, 19] and estimate how well inferred latent regions capture the human[5] performing the action. Given a video, for each frame we compute the overlap score between the latent region and the ground truth bounding box of the human. The overlap $O(b_k^j, b_{gt}^j)$ is defined by the "intersection-over-union" metric between inferred and ground truth bounding box. The total localization score per video is computed as an average of the overlap scores of the frames: $\frac{1}{T} \sum_{j=1}^{T} O(b_k^j, b_{gt}^j)$. Note, since our latent regions may not span the entire video, instead of dividing by the number of frames $T$, we divide by the total length of the inferred latent regions. To be consistent with the literature [9, 19], we calculate the localization score of each test video given its ground truth action label.

Table 1 illustrates average localization scores[6]. It is clear that our model with **Context** achieves considerably better localization than without (**Region**) especially with two latent regions. This can be explained by the fact that in UCF-Sports background tends to be discriminative for classification; hence without proper context a latent region is likely to drift to the background (which reduces localization score). Context in our model models the background and leaves the latent regions free to select perceptually salient regions of the video. Numerically, our full model (**Region+Context**) outperforms the model of Lan *et al.* [9] (despite [9] having person detections and actor annotations

| | Region | | Region+Context | |
|---|---|---|---|---|
| | $K=1$ | $K=2$ | $K=1$ | $K=2$ |
| Ave. | 60.6 | 47.6 | 68.5 | 63.8 |

| | Region, $K=1$ | | Region+Context, $K=1$ | |
|---|---|---|---|---|
| | Corr. | $\chi^2$ | Corr. | $\chi^2$ |
| Ours | 0.36 | 1.64 | 0.36 | 1.55 |
| [11] | 0.44 | 1.43 | 0.46 | 1.31 |

Table 2: **Average amount of gaze (left):** Table shows fraction of ground truth gaze captured by the latent region(s) on test videos; context improves the performance. **Top-down saliency prediction (right):** $\chi^2$ distance and norm. cross-correlation between predicted and ground-truth gaze densities.

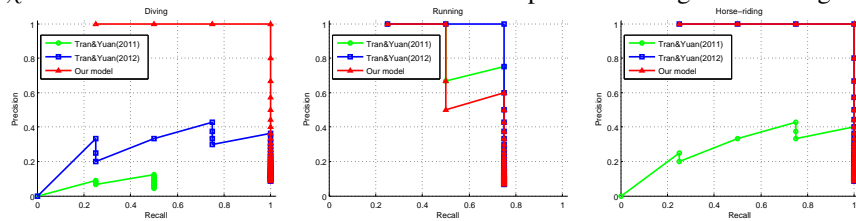

Figure 2: **Precision-Recall curves for localization:** We compare our model (**Region+Context** with K=1 latent region) to the method from [18] and [19].

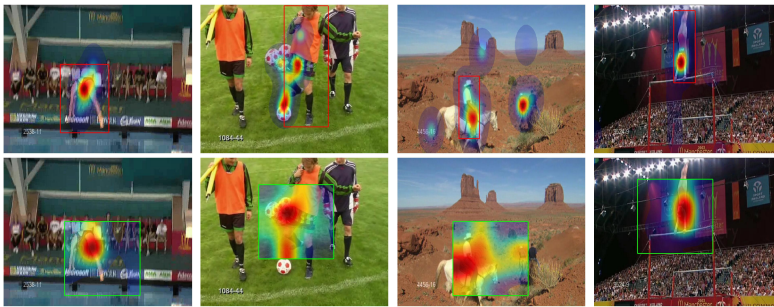

Figure 3: **Localization and gaze prediction:** *First row*: groundtruth gaze and person bounding box, *second row*: predicted gaze and extent of the latent region in the frame.

at training). We cannot compare our average performance to Tran and Yuan [19] since their approach is evaluated only on 3 action classes out of 10, but we provide their numbers in Table 1 for reference. We build Precision-Recall (PR) curves for our model (**Region+Context**) and results reported in [19] to better evaluate our method with respect to [19] (see Fig. 2). We refer to [19] for experimental setup and evaluate the PR curves at $\sigma = 0.2$. For the 3 classes in [19] our model performs considerably better for "diving" action, similarly for "horse-riding", and marginally worse for the "running".

**Gaze localization and prediction:** Since our model is driven by eye-gaze, we also measure how much gaze our latent regions can actually capture on the test set and whether we can predict eye-gaze saliency maps for the inferred latent regions. Evaluation of the gaze localization is performed in a similar fashion to the evaluation of action localization described earlier. We estimate amount of gaze that falls into each bounding box of the latent region, and then average the gaze amount over the length of all the latent regions of the video. Thus, each video has a gaze localization score $s_g \in [0, 1]$. Table 2 (left) summarizes average gaze localization for different variants of our model. Noteworthy, we are able to capture around $60\%$ of gaze by latent regions when modeling context. We estimate gaze saliency, as described in Sect. 4. Qualitative results of the gaze prediction are illustrated in Fig. 3. For quantitative comparison we compute normalized cross-correlation and $\chi^2$ distance between predicted and ground truth gaze, see Table 2 (right). We also evaluate performance of bottom-up gaze prediction [11] within inferred latent regions. Better results of bottom-up approach can be explained by superior low-level features used for learning [11]. Still, we can observe that for both approaches the full model (**Region+Context**) is more consistent with gaze prediction.

# 6 Conclusion

We propose a novel weakly-supervised structured learning approach for recognition and spatio-temporal localization of actions in video. Special case of our model with two temporally ordered paths and context can be solved in linear time complexity. In addition, our approach does not require actor annotations for training. Instead we rely on gaze data for weak supervision, by incorporating it into our structured loss. Further, we show how our model can be used to predict top-down saliency in the form of gaze density maps. In the future, we plan to explore the benefits of searching over region scale and focus on more complex spatio-temporal relationships between latent regions.

## Footnotes

[1]We assume no gaze data is available for test videos.

[2]Similar observations have been made in object detection domain [25], where purely bottom-up saliency has been shown to produce responses on textured portions of the background, outside of object of interest.

[3] The feasible positions of the bounding box in a frame are constrained by its location in the previous frame.

[4]Parameter $r$ of the loss (Sect. 3) modulates importance of gaze localization within the latent region.

[5]Note that by definition the task of localizing a human is unnatural for our model since it captures perceptually salient fixed sized discriminate regions for action classification, not human localization. This unfavorably biases localization results agains our model; see Fig. 3 for visual comparison between annotated *person* regions and our inferred discriminative salient latent regions.

[6]It is worth mentioning that [19] and [9] have regions detected at different subsets of frames; thus in terms of localization, these methods are not directly comparable.

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
