[Reviews · NeurIPS 2013]

Submitted by Assigned_Reviewer_6

This paper proposes a method for action detection (localization and classification of actions) using weakly supervised information (action labels + eye gaze information, no explicit definition of bounding boxes). Overall, the spatio-temporal search (a huge spatio-temporal space) is done using dynamic programming and a max-path algorithm. Gaze information is introduced into framework through a loss which acounts for gaze density at a given location.

QUALITY: The paper seems technically sound and makes for a nice study given gaze information. An interesting extension would be the application of general gaze prediction (rather than gaze points from the same dataset). The experimental case which is still not clear is whether or not having bounding box information is more or less beneficial than gaze information. As it requires a fair bit of effort to collect both types of information on the dataset, a nice baseline to have would have been the use of the ground truth bounding boxes in the training rather than the gaze information.

CLARITY: The paper is reasonably written, though some definitions of terminology and variables are missing.
(1) In line 75, what is meant exactly by top-down saliency? This is not explained until much later in the text (around line 95).
(2) In equation 1, what is w?
(3) In line 166 - what is the variable k?

ORIGINALITY
Reasonably original and is a nice way of integrating gaze information into the action recognition problem.

SIGNIFICANCE
The introduction of gaze information is becoming more common in computer vision.


MINOR PROBLEMS
Line 70, 71 – miss-classification, miss-alignment should be misclassification and misalignment
Line 134 – further extended
Summary: Reasonably written paper which makes use of gaze information. The writing could use a bit of polishing before final publication.

Submitted by Assigned_Reviewer_7

The paper targets action recognition and localization in video. The main novelty is the use of human gaze data as supervision for learning action localization. The method is formulated within a structured-output learning framework. The loss function combines penalties for both incorrect action classification and localization. Correct action localization is assumed to coincide with the location of the gaze provided by human subjects for the training videos. Experiments for action classification and localization are reported for UCF sports dataset.

Positive:
- The use of gaze data as supervision for learning action localization is appealing since manual video annotation is very time consuming.
- The paper is clearly written and well-structured. The formulation of the method seems to be fine.
- Experimentally, action classification seems to benefit significantly from the gaze information incorporated at the training.

Negative:
- The hypothesis that gaze locations always coincides with locations of action does not seem to be verified. What's the performance of action localization by gaze on the test set? Will gaze data be reliable in all cases? What if training videos contain multiple actions?
- The output of an automatic person detector/tracker could be thought of as an alternative cue for learning action localization. Compared to gaze, person detection does not require any manual intervention. It seems your method could be easily adapted to incorporate person likelihood maps (produced by a person detector) instead of the gaze map g. It would be good to see comparison of both.
- Weakly-supervised action localization has been explored without gaze e.g. in [Siva and Xiang, BMVC11] (missing reference). This work should be discussed and compared experimentally, if possible.
- Action classification is reported for a non-standard experimental setup (l.313). To enable comparison to most of the methods reporting for UCF-Sports, results for the standard 10-fold cross-validation setup should be reported.
- Evaluation of action localization seems to be done for frames where action is detected but not for the entire duration of the action (l.365-366). If this is the case, the evaluation does not penalize low recall and the localization results in Table 1 are probably non-comparable to other methods.

Detailed comments:
- since y={-1,1}, "otherwise" value in eq. (4) should probably be just 1.
Summary: Summary: I like the idea of using gaze as supervision for localization, the gaze data is well-integrated into the structured-output learning framework. Due to negative points above, however, the paper is not as conclusive as it could be.

Submitted by Assigned_Reviewer_8

This paper proposes the use of eye-gaze data for spatio-temporal action localization in videos, showing improved action classification accuracy and state-of-the-art action localization in videos. The method of incorporating this information is a latent SVM framework with modified constraints. The authors conduct various experiments to show that their method performs better than existing methods for localization and using eye gaze data could serve as a better prior than bounding boxes as done by previous works.

Overall, I found the paper to be well written but the experimental results a little lacking. The concerns are explained below.

Pros:
- Interesting problem formulation of using eye-gaze data instead of bounding box data for learning
- Experimental results are thorough and various problem settings are evaluated such as action classification, localization and gaze prediction. The experimental results suggest that the method proposed in the paper outperforms existing methods on the given tasks.

Cons:
- The cost and difficulty of collecting eye-gaze data is not well justified by the observed improvements. While eye-gaze data may be more natural in some ways, it is significantly harder to collect as compared to bounding box annotations which can be easily collected via crowdsourcing as no specialized equipment is required. This makes it significantly harder to scale this approach to other datasets.
- How important is it to model gaze in a category specific manner? If gaze could be used for general saliency prediction, then the same method could be applied to other datasets more easily.
- It is not clear why the authors do not compare their results to other existing methods for action classification such as [Z1] and [Z2] (mentioned below). The performance achieved by them is 85.6% and 86.5% as compared to the reported performance of 82.1%. It should be noted that [Z1] and [Z2] do not report results on action localization but the same can be said for [16] and [12] which are reported here. While [Z1] and [Z2] use chi-square kernels for learning, it is an advantage of their methods and it is not clear if the same set of kernels can be applied in the formulation presented here.
- What is the corresponding localization result of the proposed method on the 3 categories used by Tran and Yuan[18]? It would be nice to include this in the figure caption.
- There are no baselines provided for gaze localization and prediction. It is possible to use some simple methods such as exemplar-based detectors[Z3] for gaze prediction to show the necessity/strength of your method.

References (I use Z to be distinct from paper references)
[Z1] Evaluation of local spatio-temporal features for action recognition. In BMVC, 2010.
[Z2] Learning hierarchical invariant spatio-temporal features for action recognition
with independent subspace analysis. In CVPR, 2011.
[Z3] Ensemble of Exemplar-SVMs for Object Detection and Beyond. In ICCV, 2011.

The paper is generally well-written but here are some minor language errors:
- L51: this comes at expense --> add "the"
- L69-70: miss --> mis
- L82: proved --> have proven
- L134: extend --> extended
Summary: This paper proposes a modification of latent SVM to incorporate eye-gaze information for doing action classification and localization. While the results, are promising, there are some concerns stated in the previous question that should be addressed by the authors.
Author Feedback

Author rebuttal: We would like to thank reviewers for their helpful and thoughtful comments.

R1
==
We will address the specific comments improving the clarity of the paper.

R2
==
Q1:The hypotheses gaze locations always coincides with locations of actions does not seem to be verified...Will gaze data be reliable in all cases?

There is no guarantee that the eye-gaze data will always coincide with the region of interest for the action (we illustrate this in Fig. 1 (b)). However, portions of eye-gaze fixations usually do fall within location of an action at some point within action execution. This is precisely the reason why we treat eye-gaze as weak supervisory signal and do not aggressively penalize regions that do not contain eye-gaze during training (see Eq. 5 and 6). Moreover, we experimentally show that our localization results (Fig. 2, action localization reported on the test set) outperform other methods that use combination of hand annotations and person detections.

Q2:What if training videos contain multiple actions?

Since we are localizing actions in space and time, presence of multiple actions in the training/test videos can also be handled with the proposed framework. The fact that our model only assumes that a fraction of eye-gaze fixations fall within an action region of interest (Eq. 5 and 6) further helps with that.

Q3:The output of an automatic person detector/tracker could be thought of as an alternative ... for learning action localization. ... Weakly-supervised action localization has been explored without gaze ... in [Siva & Xiang, BMVC11]

We compare our work to [9] and [16]; in [9], the action model is driven by automatic person detector in both train and test stages; and [16] explores weakly supervised action learning and is similar in nature to [Siva & Xiang, BMVC11] (which we will cite and discuss). Our model outperforms both approaches as we show in the paper. However, we acknowledge the recommendation and agree that employing human detector to replace gaze map in our model, specifically, is an interesting experiment to conduct in the future.

Q4:Action classification is reported for a non-standard experimental setup ... 10-fold cross-validation setup should be reported

For this dataset (UCF-Sports) the 10-fold or Leave-One-Out cross-validation is biased and the model tends to learn the scene of the action rather than action itself (many actions are captured in the same location, please see [9] for more details). Therefore, we use the train/test split provided in [9], which serves a more “fair” evaluation. In addition, recent works on this dataset use the same train/test splits [9,12,16,18]. The use of the train/test splits provided in [9] further allows us to compare our results more directly to the two methods that are most closely related to the proposed model - [9] and [18].

Q5:Evaluation of action localization seems to be done for frames where action is detected ... results on Table 1 are probably non-comparable to other methods.

In footnote 7 (page 7), we mention that all methods (ours, [9], and [18]) are evaluated on different subsets of frames, and therefore are not directly comparable. Because all these methods localize actions in space and time (and hence may and do fire on different subsets of frames) a perfectly direct comparison is difficult.

R3
==
Q1:The cost and difficulty of collecting eye-gaze data is not well justified ... bounding-box annotations which can be easily collected via crowdsourcing as no specialized equipment is required.

While currently the cost of eye-trackers is more expensive, we believe that collecting eye-gaze data will become more feasible in the future and it might be easier for users to watch the video in a natural setting rather than perform frame-by-frame bounding box annotation. In addition, it is not a trivial task to indicate strict spatio-temporal boundaries of the action. E.g., it is an open question whether the bounding box of action "riding a horse" should include horse or not. Eye-gaze circumvents such labeling choices. Moreover, our approach shows superior performance over the methods [9, 12, 18] that were trained with bounding-box hand annotation.

Q2:It is not clear why the authors do not compare their results to ... [Z1] and [Z2]

We would like to point out that both [Z1] and [Z2] evaluate their approach based on the Leave-One-Out cross-validation, which is different from our evaluation which is based on train/test data split [9]. As it is shown in [9], LOO framework tends to memorize the scene of the action rather than learn the action model. Such a behavior leads to higher numerical results in action classification compared to the train/test split based evaluation. Due to such high scene correlation between the actions, we believe that using train/test split for learning and evaluating the method is more “fair” than using Leave-One-Out setup.

Q3:[Z1] and [Z2] use chi-squared kernels for learning, ... not clear the same set of kernels can be applied in the formulation presented here.

For simplicity and in order to compare results with [9], we employ linear kernel for learning. However, our model can be easily extended and include chi-square kernel for learning, e.g. by using approximate kernels as it is done in [12].

Q4: “What is the corresponding localization results of the proposed method on the 3 categories used by Tran and Yuan [18]?”

Average Precision (Fig. 2):
Diving:
Ours: 100; Tran & Yuan [17]: 10; Tran & Yuan [18]: 41.
Running:
Ours: 67; Tran & Yuan [17]: 70; Tran & Yuan [18]: 77.
Horse Riding:
Ours: 100; Tran & Yuan [17]: 42; Tran & Yuan [18]: 100.

Q5: “.. no baselines provided for gaze localization and prediction.”

For the final version of the paper we will compare our results to the bottom-up HoG-MBH saliency detector made publicly available a few weeks ago by S. Mathe and C. Sminchisescu [11].